# Distinction of *Alternaria* Sect. *Pseudoalternaria* Strains among Other *Alternaria* Fungi from Cereals

**DOI:** 10.3390/jof8050423

**Published:** 2022-04-20

**Authors:** Philipp B. Gannibal, Aleksandra S. Orina, Galina P. Kononenko, Aleksey A. Burkin

**Affiliations:** 1Laboratory of Mycology and Phytopathology, All-Russian Institute of Plant Protection, 196608 St. Petersburg, Russia; orina-alex@yandex.ru; 2Mycotoxicology Laboratory, All-Russia Research Institute of Veterinary Sanitation, Hygiene, and Ecology-Skryabin and Kovalenko Federal Scientific Center, All-Russia Research Institute of Experimental Veterinary Medicine, 123022 Moscow, Russia; kononenkogp@mail.ru (G.P.K.); aaburkin@mail.ru (A.A.B.)

**Keywords:** *Alternaria avenicola*, identification, barley, wheat

## Abstract

Species of the genus *Alternaria* are ubiquitous and frequently isolated from various plants, including crops. There are two phylogenetically and morphologically close *Alternaria* sections: the relatively well-known *Infectoriae* and the rarely mentioned *Pseudoalternaria*. Currently, the latter includes at least seven species that are less studied and sometimes misidentified. To perform precise identification, two primers (APsF and APsR) were designed and a sect. *Pseudoalternaria*-specific PCR method was developed. Thirty-five Russian *A*. *infectoria*-like strains were then examined. Five strains were found to be the members of the sect. *Pseudoalternaria*. Additionally, specificity of the previously developed primer set (Ain3F and Ain4R) was checked. It was found to be highly specific for sect. *Infectoriae* and did not amplify sect. *Pseudoalternaria* DNA. Identification of strains of the sect. *Pseudoalternaria* was supported and refined by phylogenetic reconstruction based on analysis of two loci, the glyceraldehyde-3-phosphate dehydrogenase gene (*gpd*), and the plasma membrane ATPase gene (*ATP*). These fungi belonged to *Alternaria kordkuyana* and *A. rosae*, which were the first detection of those taxa for the Eastern Europe. *Alternaria kordkuyana* was isolated from cereal seeds and eleuthero leaves. *Alternaria rosae* was obtained from oat seed. All strains of sect. *Pseudoalternaria* were not able to produce alternariol mycotoxin, as well as the majority of *A.* sect. *Infectoriae* strains.

## 1. Introduction

*Alternaria* Nees fungi are ubiquitous and frequently isolated from plants, soil, air, dust, and water-damaged buildings [1,2]. Many *Alternaria* species infect crop plants in the field and cause diseases, leading to significant economic losses [1,3,4].

This genus is characterized by dark colored, multicelled conidia with transverse and longitudinal septa. Conidia commonly occur in chains or sometimes remain solitary and usually contain an apical beak or tapering apical cells [5]. Early *Alternaria* taxonomy was inconsistent as it was based only on general morphological characteristics. The later taxon criteria within this genus used many morphological characteristics of conidia and a three-dimensional sporulation pattern. It led to the description of several *Alternaria* species groups [6,7,8]. Later based on combination of morphological and multilocus phylogenetic characteristics, many *Alternaria* species-groups were converted into the sections, the taxa of the sub-generic rank [9]. Among them, the *A. infectoria* species group was distanced from other small-spored *Alternaria* spp. by morphological and biochemical characters and molecular markers [10,11,12]. The majority of the *A. infectoria* species group representatives was placed in the sect. *Infectoriae*. However, some *A. infectoria*-like strains were recently placed in the sect. *Pseudoalternaria*, which is strongly supported as the sister group to sect. *Infectoriae* [13]. Currently, this section includes at least seven species.

Species of *A.* sect. *Pseudoalternaria* form primary conidiophores aggregated on agar surface or developing from aerial hyphae, simple or branched with single apical pore [14]. Conidia are relatively small (usually no larger than 32 × 10 μm), ellipsoid to obclavate, medium brown to golden brown, and mostly combined in short chains. Conidia form 3–4 transverse and 1–2 longitudinal septa. Sometimes, conidia produce short to long, simple to multi-geniculate secondary conidiophores, obtaining one to many conidiogenous loci.

There is a limited number of reports for *A.* sect. *Pseudoalternaria* fungi; however, they were isolated from various host plants in geographically distant locations. *Alternaria arrhenatheri* D.P. Lawr., Rotondo, and Gannibal, the type species of section, was isolated from: *Arrhenatherum elatius* in the USA [13]; *A. rosae* E.G. Simmons and C.F. Hill from the stem of sweet briar in New Zealand [15]; *A. parvicaespitosa* Gannibal and D.P. Lawr. from blueberry fruit in the USA [14,16]; *A. kordkuyana* Poursafar, Gannibal, Ghosta, Javan-Nikkhah, and D.P. Lawr. and *A. ershadii* A. Poursafar, Y. Ghosta and M. Javan-Nikkhah from wheat plants with black head mold symptoms in Iran [17,18]; *A. altcampina* Iturrieta-González, Dania García, and Gené and *A. inflata* Iturrieta-González, Dania García, and Gené from herbivore dung in Spain [19]. The fungus *A. brassicifolii* found on napa cabbage in Korea was also described as a species of *A.* sect. *Pseudoalternaria* [20]; however, it stood out from all other species on their phylogenetic tree and its section relation is not clear [18]. The representatives of sect. *Pseudoalternaria* can be found widely on different substrates. However, information on genetic diversity, distribution, and abundance are far from complete due to the difficulty of morphological identification. Similarly, there is no information on the presence of these fungi in Russia. Therefore, the biochemistry, ecology, and importance of sect. *Pseudoalternaria* remain to be clarified.

The accurate identification of *A.* sect. *Pseudoalternaria* spp. is possible only using phylogenetic analysis of several loci sequences. Internal transcribed spacer ITS, *ATP*, and *gpd* genes are most informative and usable for these fungi at this time [17,19,20]. There are no primers for detection through specific PCR. The specific primers have been previously developed to identify *A. infectoria*-like strains [21], but their specificity needs to be verified in the context of the modern understanding of the *Alternaria* division into sections.

*Alternaria* fungi are producers of a variety of secondary metabolites, some of which may be phytotoxins or mycotoxins [22,23]. *Alternaria* mycotoxins are widely found in a variety of food and feed and affect the health of consumers [24]. The major mycotoxins produced by these fungi are alternariol (AOH), alternariol monomethyl ether, altenuene, altertoxins, and tenuazonic acid [4,25,26]. AOH is one of most often analyzed and detected [27,28,29,30,31] and have genotoxic, mutagenic, and carcinogenic effects in humans and animals [24].

The mycoxin-producing ability of *Alternaria* spp. strains in sect. *Alternaria* and *Infectoriae* might differ significantly [26,32,33,34]. Strains in sect. *Alternaria* usually produce some or all of the mycotoxins listed above, and sometimes the quantity of synthetized metabolites can be high. Culture extracts of sect. *Infectoriae* strains are usually non-toxic or have low toxicity [32]. At the same time, the toxin-producing ability of *A.* sect. *Pseudoalternaria* fungi has not yet been studied.

The aims of this study were to incontrovertibly identify Russian *A*. sect. *Pseudoalternaria* strains by molecular phylogenetic, PCR, and morphological analyses and to define their ability to produce the alternariol mycotoxin.

## 2. Materials and Methods

### 2.1. Alternaria Strains

Sixty-six strains of *Alternaria* spp. from the collection of the Laboratory of Mycology and Phytopathology of the All-Russian Institute of Plant Protection (St. Petersburg, Russia) were included in the study (Table 1). Of these, 35 *Alternaria* strains, mainly from European Russia, were preliminarily determined on the basis of the sum of macro- and micromorphological characters as small-spored to be *A. infectoria*-like taxa that could actually belong to sects. *Infectoriae* or *Preudoalternaria*. The other 31 strains comprised 24 species in 10 sections with one monophyletic lineage and accurate species affiliation. These strains were used to test the specificity of PCR primers developed in this study.

### 2.2. PCR Primer Development

The primer pair APsF (CCGCCGCCAATCCAGTTC) and APsR (AAGGTTGGTCTTCTCGGAAG) specific for DNA of *Alternaria* sect. *Pseudoalternaria* fungi was designed based on ATPase gene sequences from *Alternaria* spp. available in the GenBank database (Table 2). Primer design was performed using online software Primer3Plus [35]. These primers were expected to amplify a region of 424 bp.

*Alternaria* spp. were cultured on potato sucrose agar medium (PSA) for 7 days. The genomic DNA from the mycelium (10–50 mg per strain) was isolated using a genomic DNA purification kit (Thermo Fisher Scientific, Vilnius, Lithuania) according to the manufacturer’s protocol.

The specificity of the primers APsF and APsR was analyzed by PCR with DNA, using all 66 *Alternaria* spp. strains. The amplification was also performed with the primers Ain3F and Ain4R that had previously been developed for the identification of the *A. infectoria*-like fungi [21]. Amplification was done in a 25 µL reaction mix comprising 1 × PCR-buffer with 25 mM MgCl_2_, 0.2 mM dNTP, 0.5 U Taq polimerase (all reagents Thermo Fisher Scientific, Vilnius, Lithuania), and 0.5 µM primers (Eurogen, Moscow, Russia) with a BioRad C1000 Touch Thermal Cycler (Bio-Rad Laboratories, Hercules, CA, USA) using the following cycling protocol: 95 °C for 3 min, 40 cycles of 95 °C for 20 s, 65 °C for 20 s, and 72 °C for 40 s, followed by final elongation of 72 °C for 3 min. Visualization of results was performed by electrophoresis of amplification products in 1% agarose gel.

### 2.3. Genomic DNA Isolation, Sequencing, and Phylogenetic Analysis

The *Alternaria* strains whose DNA was amplified with the primers APsF and APsR were included in the phylogenetic study. The region of the glyceraldehyde-3-phosphate dehydrogenase gene (*gpd*) was amplified using primers gpd1 and gpd2 [38]. Primers ATPDF1 and ATPDR1 were used to amplify part of plasma membrane ATPase gene (*ATP*) [9]. Sequencing of the fragments was done on an ABI Prism 3500 sequencer (Applied Biosystems, Hitachi, Japan) using a BigDye Terminator v3.1 cycle sequencing kit (Applied Biosystems, Foster City, CA, USA).

Alignment of the sequences obtained for each strain was performed using Mega X 10.1 program [39]. Basic Local Alignment Search Tool (BLAST) was used to perform similarity search, by comparing the consensus sequences with sequences in NCBI GenBank database. The closest matching sequences were added to the alignment (Table 2). Phylogenetic analysis of combined sequences consisted of maximum likelihood (ML) and maximum parsimony (MP) performed with Mega X 10.1. ML analysis was completed on a neighbor joining starting tree, generated automatically. Nearest neighbor interchange was used as the heuristic method for tree inference. The best nucleotide substitution model used for building the ML trees (TN93 + G) was also determined in MEGA X 10.1. MP analysis was performed using the heuristic search option with 100 random taxon additions and the subtree pruning regrafting method as the branch-swapping algorithm. All characters were unordered and of equal weight, and gaps were treated as missing values. Input parameter “maxtrees” was set to 100, and branches of zero length were collapsed. Bootstrap supports values for ML and MP trees branches were calculated with 1000 replicates. Additionally, Bayesian probability (BP) calculation was done with Mr. Bayes v. 3.2.1. in Armadillo v. 1.1 [40]. using a Markov chain Monte Carlo (MCMC) sampling method. The general time-reversible model of evolution, including estimation of invariable sites and assuming a gamma distribution with six rate categories, was used for Bayesian inference analyses. Four MCMC chains were run simultaneously, starting from random trees for 1000 generations and sampled every tenth generation for a total of 10,000 trees. Sequence data was deposited in GenBank (MW478365-MW478374).

### 2.4. Morphology Characterization

For examination of colony morphology, the strains were grown on potato carrot agar (PCA) [15] with 12:12 h L:D photoperiod for 7 days and on PSA without lighting. For sporulation assessment, the strains were cultured on PCA at 24 °C with 12:12 h L:D photoperiod for 3–5 days [15]. Observations of conidiation were made with a SZX16 stereomicroscope and BX53 microscope (Olympus, Tokyo, Japan). Images were captured with a Prokyon camera (Jenoptik, Jena, Germany).

### 2.5. Toxin-Producing Ability

For analysis of toxin-producing ability, 25 *Alternaria* strains belonging to four sections (14 sect. *Infectoriae,* 5 sect. *Pseudoalternaria*, and 3 sects. *Alternaria* and *Panax*) were selected. The selected strain was grown in 15-mL glass vials containing 1.0 mL malt extract agar (MEA), 1 g of polished rice (rice), or pearl barley (barley) in 3–6 replicates. The moisture content of the rice and barley was adjusted by the addition of 2 mL of water before autoclaving at 120 °C for 20 min. Inoculation of MEA and groats was done by suspension of conidia (50 µL at 50 CFU/µL). The strains were cultured for 7 days at 25 °C in complete darkness.

For mycotoxin extraction, 3 mL of the mixture acetonitrile and water (84:16, *v*/*v*) was added in each vial. The vials ware shaken vigorously and left to incubate for 12–14 h. After repeated shaking and tenfold dilution with a buffer solution, the extract was used for indirect competitive enzyme-linked immunosorbent assay [41]. If necessary, 100-, 1000- or 10,000-fold dilutions of the extract were made with a buffer solution containing 10% of the acetonitrile and water mixture (84:16, *v*/*v*). Determination of AOH with a detection limit of 0.4 ng/mL was performed using certified kits and the manufacturer’s protocol (VNIIVSGE, Moscow, Russia). The toxin-producing ability of the *Alternaria* spp. strains was determined as the AOH content in 1 g of substrate (µg/g).

### 2.6. Statistical Analysis

Experiments to study the toxin-producing ability of the strains were conducted twice, using at least three biological replicates. Statistical analysis of the obtained results was performed using Statistica 10.0 software. The mean values and the confidence interval were calculated at a 95% significance level.

## 3. Results

### 3.1. Specificity of Primers and Strains Identification with PCR

PCR with primers APsF and APsR amplified DNA of only both representative strains of sects. *Pseudoalternaria*, *A. arrhenatheri*, and *A. parvicaespitosa* (Table 1). However, with primers Ain3F and Ain4R, DNA was amplified for all seven representative *Alternaria* sect. *Infectoriae* strains. Importantly, PCR with both primers sets did not amplify DNA of the 21 *Alternaria* spp. belonging to the eight other *Alternaria* sections and *A. brassicae*.

The newly designed primers APsF and APsR, along with Ain3F and Ain4R, were used to search for *Alternaria* sect. *Pseudoalternaria* among 35 local *A. infectoria*-like strains. These strains, according to the sum of morphological characters, could be representatives of both sects. *Infectoriae* and *Pseudoalternaria*. DNA from 30 *Alternaria* strains was amplified with primers Ain3F and Ain4R, but five other *Alternaria* spp. were amplified with primers APsF and APsR. The latter five strains were included in the further phylogenetic study.

### 3.2. Molecular Phylogeny

Adjusted and aligned *gpd* and *ATP* sequences had the lengths of 492 and 1232 bp with 82 (16.7%) and 185 (15.0%) parsimony-informative sites per genome locus, respectively. Topology of trees built by different methods was the same and also was concordant with phylogenetic relation between *Alternaria* species reconstructed previously [2,17,18]. Four local *Alternaria* strains formed a compact clade with high bootstrap support (ML/MP/BP 99/99/1.0) containing *A. kordkuyana* IRAN 16888 (Figure 1). One local *Alternaria* strain MF P457041 clustered with *A. rosae* EGS 41-130 with bootstrap support ML/MP/BP 98/-/0.96. MF P457041 and CBS 121341, T differed by four substitutions in the *ATP* gene, two points in the non-coding region, one synonymous substitution, and one point mutation.

### 3.3. Alternaria Sect. Pseudoalternaria Morphology

Colonies of *A. kordkuyana* MF P094121 on PSA were 79 mm diam. after 7 days without lighting, flat, felty, light gray at center, and white at edge, with light taupe reverse; on PCA, 79 mm diam. after 7 days with 12:12 h L:D photoperiod, flat, velvet, with poor aerial mycelium, light brown on top, and reverse (Figure 2C). Conidia were ellipsoidal, obclavate, smooth, or verrucose, usually with darkened and constricted median transeptum, 9–34 × 6–12 μm (av. 20 × 9 μm), mainly with secondary conidiophores from apical cell 4–22 × 4–5 μm (av. 10 × 4 μm) (Figure 2A). Conidia were formed in simple or branched chains up to five units in a row (Figure 2B). Sporulation clamps consisted initially of 2–7 conidia or up to 15 (20) conidia in the old cultures. Branching of the chains occurred mainly due to the presence of 2–4 conidiogenous loci on the apical secondary conidiophores. Lateral secondary conidiophores were not observed. In general, the three-dimensional sporulation pattern was similar to that of sect. *Infectoriae* but differed by shorter chains and smaller conidial clumps.

Colonies of *A. rosae* MF P457041 on PSA were 71 mm diam., flat, felty, dark gray at center, and white at edge, with dark brown reverse, turning light brown towards the edge; on PCA, 76 mm diam., flat, cottony, white, and taupe in center and light brown at edge, with light brown reverse (Figure 2D). This strain in our study remained sterile on PCA and PSA under the used temperature and lighting conditions.

### 3.4. Alternariol-Producing Ability

Only four of the 25 *Alternaria* spp. strains analyzed produced AOH with concentrations ranging from 1 to 1910 µg/g, depending on the substrate (Table 1). Of these, three strains of *Alternaria* sect. *Alternaria*, as well as MF P457121 in sect. *Infectoriae*, were AOH-producing. The remainder of the representatives of sect. *Infectoriae* (13 strains), all strains belonging to sect. *Pseudoalternaria* (5 strains) and *Panax* (3 strains), did not produce the AOH on the analyzed substrates in detectable amounts. 

## 4. Discussion

Accurate identification of *Alternaria* spp. is complicated and it is often only possible using molecular methods [2]. The phylogenetic approach to differentiation of *Alternaria* spp. requires the analysis of sequences of several loci [2], while specific PCR is simple and convenient when primer specificity is sufficiently precise. A number of primers specific for *Alternaria* were designed. There are primers for identification of *Alternaria* fungi at the genus level [42]. Primer sets intended for identification of *Alternaria alternata*-like fungi [43,44,45] have specificity that is limited to sect. *Alternaria*. One of these primer sets was successfully used for quantitative detection of sect. *Alternaria* fungi in cereals grain [46]. The primers Ain3F and Ain4R were developed for the identification of the *A. infectoria*-like fungi [21]. Their specificity for sect. *Infectoriae* was re-examined and confirmed in this study. The present study designed and validated the first primer set (APsF and APsR) specific for *Alternaria* sect. *Pseudoalternaria* fungi. These primers, in combination with primer sets for sects. *Alternaria* and *Infectoriae*, can be used for analysis of mycobiota of grain or other agricultural or clinical samples. Sometimes, grain samples with high *Alternaria* infection levels should be assessed for their potential toxicity. With these three primer sets, PCR can be used for this purpose as a simple and quick method.

In the present study, of 35 Eastern European *A. infectoria*-like strains, five strains (14.3%) were found to belong to sect. *Preudoalternaria.* All of these stains came from Leningrad and Belgorod Regions of European Russia. Despite the fact that *A.* sect. *Pseudoalternaria* fungi have previously been found on common crops, wheat [17,18], apple [47], and napa cabbage [20], no *A.* sect. *Pseudoalternaria* fungi have been detected in Eastern Europe. The present study is the first to establish the presence of *A.* sect. *Pseudoalternaria* species in Russia.

Four local strains were identified as *A. kordkuyana* and were isolated from wheat and oat grain (Poaceae) and eleuthero leaves (Siberian ginseng, Araliaceae) in central and northwestern European Russia. Previously, *A. kordkuyana* was isolated from wheat plant in Iran [17], from apple in Chili [47], and from herbivore dung in Spain [19]. In earlier unpublished work [48], a group of six short-chained *A. infectoria*-like strains from poaceous plants were found. All strains had ITS region sequences that differed from genuine *A. infectoria* species group strains. Five of them have identical sequences, and one of those strains, MF P457051, still exists in the collection and was used in the present study. Thus, with a high degree of confidence, we can report that *A. kordkuyana* was also found on *Triticum durum* in the Omsk Region (western Siberia) and *Hordeum vulgare* in the suburbs of St. Petersburg. Another strain of an unidentified species of *A.* sect. *Pseudoalternaria* was isolated from *Leymus arenarius* in the Murmansk Region (Northwestern European Russia).

One local strain isolated from oat grain in the present study was presumptively identified as *A. rosae.* Information on the distribution of this species is fragmentary. Initially, *A. rosae* was isolated from stem lesions of sweet briar (*Rosa rubiginosa*, Rosaceae) in New Zealand and described by E.G. Simmons [15]. Additionally, this species was isolated from the roots of *Arabidopsis thaliana* (Brassicaceae) in Germany and its full genome was sequenced [49].

Apparently, *A. kordkuyana* and other *A.* sect. *Preudoalternaria* species are cosmopolitan and not confined to one species or family of host plants. Existing information indicates a potentially wide distribution but low occurrence of these fungi in the mycobiota of crops.

*Alternaria kordkuyana* and *A. rosae* strains had similar growth rate. The diameter of the colonies was similar to that of previously studied strains of *A. kordkuyana* [17] and *A. rosae* [15]. Conidia of *A. kordkuyana* MF P094121 were average 20 × 9 μm, mainly with oblong secondary conidiophores from apical cell av. 10 × 4 μm. According Poursafar et al. [17], conidia of this species were 15–50 × 7–12 μm, with secondary conidiophores arising from the apical cell, which may be up to 20 μm long.

Perhaps *Alternaria* fungi are abundant in the grain mycobiota of all cereal-producing countries [50,51,52,53], including Russia [46,53]. *Alternaria* sect. *Alternaria* fungi are the most frequent group among *Alternaria* spp. in grain [34,46,54,55] and are also widespread on other plants [1,56]. *Alternaria* sect. *Infectoriae* are also abundant in cereal grain and have often been reported to be on poaceous plants [10,11,17,57], but were occasionally detected on other hosts [11,58,59]. In the present study, strains of both sections were also mainly isolated from cereal plants. Presumably, *A.* sect. *Pseudoalternaria* fungi is also common in cereals, but when identified by morphological characters, may be misidentified as *Alternaria* species from sect. *Infectoriae*. In addition, *A.* sect. *Panax* fungi were identified on cereals [60,61,62] and may add confusion to species identification.

*Alternaria* fungi are producers of various mycotoxins [63]; their presence affects the quality and safety of grain-based food and feed [26,64]. Among *Alternaria* mycotoxins, AOH has genotoxic, mutagenic, and carcinogenic effects on humans and animals and is one of the most frequently analyzed toxins [24]. A draft EU Commission Recommendation on the monitoring of *Alternaria* mycotoxins in food was issued: the benchmark value of AOH in cereal-based foods for infants and children was 5 µg/kg [65].

This mycotoxin was detected in about 20% of samples of wheat, barley, and oats grain grown in Russia in 2009–2019, and its amounts varied from 5 to 675 µg/kg [66]. In another study, the analysis of grain samples grown in the Urals and West Siberia in 2017–2019 revealed a similar occurrence of AOH (27%) but smaller concentrations of this mycotoxin (2–53 µg/kg) [46]. The unequal distribution of occurrence and amounts of AOH in grain samples from different regions was noted [66]. Additionally, AOH was detected in grain samples from Europe [33,67], Asia [27,34], North and South America [68,69], and Australia [51].

*Alternaria* sect. *Alternaria* fungi are well known as producers of AOH [10,12,33,55]. In the present study, when cultured on the grain substrate (rice and pearl barley), three *A.* sect. *Alternaria* strains produced 2.5–4.8 times higher amounts of AOH than on MEA. The substantial effect of substrate on the intensity of AOH production is well known [70,71,72].

In Central and North Italy, the production AOH by *A.* sect. *Alternaria* strains isolated from wheat grain was determined to range from 1 to 5620 mg/kg and 1–8064 μg/g, respectively [33,55]. However, some *A.* sect. *Alternaria* strains did not produce AOH in vitro: 16% of strains (129 in total) did not produce mycotoxins [12]. However, it can be presumed that AOH-lacking strains were incorrectly identified and perhaps actually belonged to other sections.

Information on the ability of *A.* sect. *Infectoriae* to produce AOH is inconsistent. Previously, an analysis of 10 *A. infectoria* strains did not demonstrate any ability of these strains to produce AOH on nine different types of agar media [73]. Additionally, none of the 20 *A. infectoria* strains isolated from food crops in Argentina produced AOH when cultured on DRYES agar medium [74]. In contrast, 75% of *A. infectoria* strains from Italy produced AOH up to 223 μg/g when cultured on rice [33]. According to Oviedo et al. [12], 81% of *A. infectoria* strains analyzed produced AOH when cultured on ground rice-corn steep liquor medium at concentrations of 1.8–433 μg/g. In the study by Ramires et al. [55], all *A. infectoria* strains analyzed produced AOH in the range 0.3–20 mg/kg. The ability to produce AOH varies greatly between strains and geographic populations of *A.* sect. *Infectoriae* fungi. It is important to remember that precise species identification in this section is complicated and often *A. infectoria* is used in a broad sense and should be readily assumed to be an *Alternaria* sp. sect. *Infectoriae*.

In the present study, only one strain, MF P457121, of 14 *A.* sect. *Infectoriae* strains, produced AOH, and that was at a concentration 8–70 times less than in the average *A.* sect. *Alternaria* strains cultured on the same substrates. Additionally, when cultured on rice and pearl barley, the amount of AOH produced by MF P457121 was 10–20 times higher than on MEA. Previously, the production of AOH by *A.* sect. *Alternaria* strains (*A. alternata*, *A. arborescens*, and *A. tenuissima*) was 95 times higher than that by *A. infectoria* strains on a rice substrate [75].

None of the five *Alternaria* spp. from sect. *Pseudoalternaria* strains analyzed produced AOH when cultured on the various substrates tested. Currently, the toxigenicity of only one *A.* sect. *Pseudoalternaria* strain isolated from wheat grain has been reported [54], with the strain ITEM 17904 producing AOH on rice medium at a low concentration (1.5 mg/kg). There is no available information on the toxigenicity of *A. avenicola* or other *Alternaria* spp. from sect. *Panax*. In the present study, none of the three strains in sect. *Panax* produced AOH.

Therefore, *Alternaria* spp. strains from sects. *Infectoriae* and *Pseudoalternaria* have the ability to produce AOH. However, the contribution of these fungi to the contamination of grain with AOH, and possibly many other more dangerous mycotoxins, has not been adequately studied compared to that of *A.* sect. *Alternaria* fungi, as these are more abundant and have AOH production potential 2–3 orders of magnitude higher.

## Figures and Tables

**Figure 1 jof-08-00423-f001:**
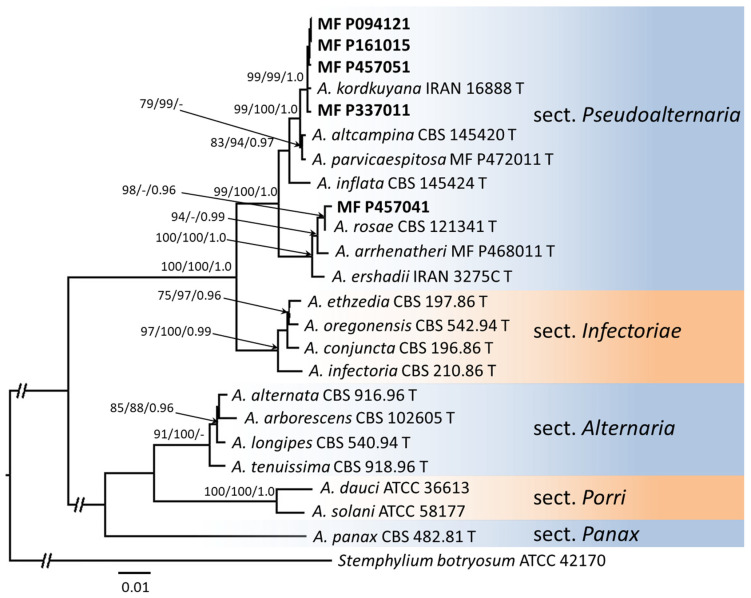
Maximum likelihood phylogenetic tree for *Alternaria* section *Pseudoalternaria* species inferred from combined *gpd* and *ATP* gene sequences. Bootstrap percentages from maximum-likelihood/maximum parsimony (>70%) and Bayesian posterior probabilities (>0.95) are given at the nodes. The strain *Stemphylium botryosum* ATCC 42170 was used as an outgroup.

**Figure 2 jof-08-00423-f002:**
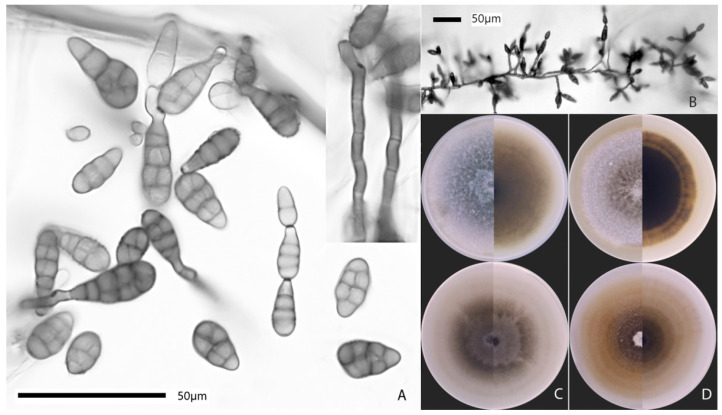
Morphology of *Alternaria* sect. *Pseudoalternaria* strains. Conidia (**A**) and conidiophores (**B**) of *A. kordkuyana* MF P094121 after 5–7 days of incubation on potato carrot agar under an alternating light/dark cycle. Colonies of *A. kordkuyana* MF P094121 (**C**) and *A. rosae* MF P457041 (**D**) on potato sucrose agar (**above**) and potato carrot agar (**below**), upper (**left**) and reverse (**right**) view.

**Table 1 jof-08-00423-t001:** *Alternaria* spp. strains used for the study and results of specific PCR and their alternariol (AOH) producing ability.

Species	Strain ID * and Status **	Host/Substrate	Origin	Results of PCR with Specific Primers		AOH Content after Growth on Different Media, µg/g		
				APsF/APsR	Ain3F/Ain4R	MEA	Barley	Rice
**Section *Pseudoalternaria***								
*A. arrhenatheri*	MF P468011, T	*Arrhenatherum elatius*	USA	+	−			
*A. kordkuyana*	MF P094121	*Triticum aestivum*, seed	Russia, Leningrad region, 2006	+	−	0	0	0
	MF P161015	*Eleutherococcus* sp., leaf	Russia, Leningrad region, 2001	+	−	0	0	0
	MF P337011	*Hordeum vulgare*, seed	Russia, Belgorod region, 2011	+	−			
	MF P457051	*Avena sativa*, seed	Russia, Leningrad region, 2003	+	−	0	0	0
*A. parvicaespitosa*	MF P472011, T	*Vaccinium corymbosum*	USA	+	−	0	0	0
*A. rosae*	MF P457041	*Avena sativa*, seed	Russia, Leningrad region, 2003	+	−	0	0	0
**Section *Infectoriae***								
*A. arbusti*	MF P527011 (EGS 91-136), T	*Pyrus* sp., leaf	USA, 1991	−	+			
*A. infectoria*	MF P492011 (CBS 210.86), T	wheat, stem	UK, 1969	−	+			
*A. metachromatica*	MF P497011 (CBS 553.94), T	wheat	Australia	−	+			
*A. oregonensis*	MF P493011 (CBS 542.94), T	wheat, leaf	USA, 1970	−	+			
*A. triticimaculans*	MF P523011 (CBS 578.94), T	wheat, leaf	Argentina, 1993	−	+			
*A. triticina*	MF P491011 (EGS 17-061), R	wheat	India, 1960	−	+			
*A. viburni*	MF P526011 (CBS 119407), T	*Viburnums* sp., stem	Netherlands, 2001	−	+			
*Alternaria* sp.	MF P022011	barley, seed	Russia, Kirov region, 2004	−	+			
	MF P026011	rapeseed, leaf	Russia, Leningrad region, 2005	−	+			
	MF P058011	barley, seed	Russia, Primorsky region, 2006	−	+	0	0	0
	MF P094101	wheat, seed	Russia, Leningrad region, 2006	−	+			
	MF P094111			−	+			
	MF P094161			−	+			
	MF P094191			−	+	0	0	0
	MF P094211			−	+			
	MF P094221			−	+	0	0	0
	MF P094251			−	+			
	MF P094261			−	+			
	MF P094301			−	+	0	0	0
	MF P094311			−	+			
	MF P094331			−	+			
	MF P185021	radish, fruit	Russia, Moscow region, 2008	−	+			
	MF P240281	cabbage, leaf	Russia, Dagestan, 2009	−	+			
	MF P266151	wheat, leaf	Russia, Krasnodar region, 2002	−	+			
	MF P276011	sunflower, leaf	Russia, Dagestan, 2009	−	+			
	MF P346061	wheat, leaf	Czech Republic, 2002	−	+			
	MF P438011	wheat, seed	Russia, Irkutsk region, 2003	−	+	0	0	0
	MF P447021	wheat, seed	Russia, Leningrad region, 2003	−	+	0	0	0
	MF P447041			−	+	0	0	0
	MF P452091			−	+	0	0	0
	MF P455031			−	+	0	0	0
	MF P457121	oat, seed	Russia, Leningrad region, 2003	−	+	3 ± 1	60 ± 20	30 ± 9
	MF P470021	rye, seed	Russia, Leningrad region, 2003	−	+	0	0	0
	MF P507031	wheat, seed	Russia, Leningrad region, 2003	−	+	0	0	0
	MF P529031	wheat, seed	Russia, North Ossetia, 2004	−	+	0	0	0
	MF P533011	wheat, seed	Russia, Krasnodar region, 2004	−	+	0	0	0
	MF P598011	potato, leaf	Russia, Kirov region, 2008	−	+			
**Section *Alternaria***								
*A. alternata*	MF P495011 (CBS 916.96), T	peanut	India	−	−			
	MF P094241	wheat, seed	Russia, Leningrad region, 2006	−	−	10 ± 4	53 ± 21	60 ± 16
	MF P455041	wheat, seed	Russia, Leningrad region, 2003	−	−	198 ± 16	1525 ± 179	1060 ± 210
*A. alternata* (*=A. longipes*)	MF P334011 (EGS 30.033), R	tobacco	USA	−	−			
*A. alternata* (*=A. tenuissima*)	MF P266071	wheat, seed	Russia, Krasnodar region, 2002	−	−	422 ± 52	1.23 ± 0.03	1910 ± 64
	MF P590011	eggplant, leaf	Russia, Moscow region, 2008	−	−			
	MF P597011	potato, leaf	Russia, Kirov region, 2008	−	−			
*A. arborescens*	MF P498011 (CBS 102605), T	tomato, stem	USA	−	−			
	MF P582011	tomato, leaf	Russia, Irkutsk region, 2008	−	−			
**Section *Brassicicola***								
*A. brassicicola*	MF P156011	cabbage, leaf	Russia, Adygeya, 2008	−	−			
**Section *Gypsophilae***								
*A. nobilis*	MF P307011 (CBS 163.63), R	carnation, leaf	1963	−	−			
**Section *Japonicae***								
*A. japonica*	MF P180011	radish, fruit	Russia, Moscow region, 2008	−	−			
**Section *Panax***								
*A. avenicola*	MF P059031	barley, seed	Russia, Leningrad region, 2006	−	−	0	0	0
	MF P071011	barley, seed	Russia, Leningrad region, 2006	−	−	0	0	0
	MF P457031	oat, seed	Russia, Leningrad region, 2003	−	−	0	0	0
*A. photistica*	MF P347011 (EGS 35-172), R	foxglove	UK, 1982	−	−			
**Section *Porri***								
*A. dauci*	MF P182011	carrot, leaf	Russia, Moscow region, 2008	−	−			
*A. linariae*	MF P580181	tomato, leaf	Russia, Irkutsk region, 2008	−	−			
*A. solani*	MF P601031	potato, leaf	Russia, Omsk region, 2008	−	−			
**Section *Radicina***								
*A. radicina*	MF P190031	carrot, leaf	Belarus, 2008	−	−			
**Section *Sonchi***								
*A. sonchi*	MF P031031	*Sonchus* sp., leaf	Russia, Krasnodar region, 2005	−	−			
**monophyletic lineage**								
*A. brassicae*	MF P165011	horseradish, leaf	Russia, Adygeya, 2008	−	−			

* Strain numbers with the acronym MF refers to the pure culture collection of the All-Russian Institute of Plant Protection (VIZR, Laboratory of Mycology and Phytopathology), St. Petersburg, Russia. EGS—personal collection of Dr. E.G. Simmons, Crawfordsville, IN, USA. CBS—culture collection of the Westerdijk Fungal Biodiversity Institute, Utrecht, The Netherlands; ** T—ex-type strain, R—representative strain.

**Table 2 jof-08-00423-t002:** *Alternaria* spp. sequences used for phylogenetic study and primer design.

Species	Strain ID * and Status **	Host/Substrate	Origin	GenBank Accessions ***		Reference
				*gpd*	*ATP*	
**Section *Alternaria***						
*A. alternata*	CBS 916.96 (EGS 34-016), T	peanut	India	AY278808	JQ671874	[13,36]
*A.arborescens*	CBS 102605 (EGS 39-128), T	tomato	USA	AY278810	JQ671880	[13,36]
*A. longipes*	CBS 540.94 (EGS 30-033), R	*Nicotiana tabacum*	USA	AY278811	JQ671864	[13,36]
*A. tenuissima*	CBS 918.96 (EGS 34-015), R	*Dianthus* *caryophyllus*	UK	AY278809	JQ671875	[13,36]
**Section *Infectoriae***						
*A. ethzedia*	CBS 197.86 (EGS 37-143), T	*Brassica napus*	Switzerland, 1981	AY278795	JQ671805	[13,36]
*A. infectoria*	CBS 210.86 (EGS 27-193), T	wheat	UK, 1969	AY278793	JQ671804	[13,36]
*A. conjuncta*	CBS 196.86 (EGS 37-139), T	*Pastinaca sativa*	Switzerland, 1982	AY562401	JQ671824	[13,36]
*A. oregonensis*	CBS 542.94 (EGS 29-194), T	*Triticum aestivum*	USA, 1970	FJ266491	JQ671827	[13,37]
**Section *Panax***						
*A. panax*	CBS 482.81 (EGS 29-180), R	*Aralia racemosa*	USA	JQ646299	JQ671846	[13]
**Section *Porri***						
*A. dauci*	ATCC 36613, R	carrot	USA	AY278803	JQ671907	[13,36]
*A. solani*	ATCC 58177, R	tomato	Mexico	AY278807	JQ671898	[13,36]
**Section *Pseudoalternaria***						
*A. altcampina*	CBS 145420, T	goat dung	Spain	LR133900	LR133906	[19]
*A. arrhenatheri*	MF P468011, T	*Arrhenatherum* *elatius*	USA	JQ693635	JQ693603	[13]
*A. ershadii*	IRAN 3275C, T	wheat, head	Iran	MK829645	MK829643	[18]
*A. inflata*	CBS 145424, T	rabbit dung	Spain	LR133938	LR133966	[19]
*A. kordkuyana*	IRAN 16888, T	wheat	Iran	MF033826	MF033860	[17]
	MF P094121	*Triticum aestivum*, seed	Russia, Leningrad region, 2006	**MW478365**	**MW478370**	
	MF P161015	*Eleutherococcus* sp., leaf	Russia, Leningrad region, 2001	**MW478366**	**MW478371**	
	MF P337011	*Hordeum vulgare*, seed	Russia, Belgorod region, 2011	**MW478367**	**MW478372**	
	MF P457051	*Avena sativa*, seed	Russia, Leningrad region, 2003	**MW478369**	**MW478374**	
*A. parvicaespitosa*	MF P472011, T	*Vaccinium* *corymbosum*	USA	MF033842	KJ908217	[14,17]
*A. rosae*	CBS 121341 (EGS 41-130), T	*Rosa rubiginosa*	New Zealand	JQ646279	JQ671803	[13]
	MF P457041	*Avena sativa*, seed	Russia, Leningrad region, 2003	**MW478368**	**MW478373**	
**outgroup**						
*Stemphylium* *botryosum*	ATCC 42170, R	*Medicago sativa*	USA	AY278820	JQ671767	[13,36]

* Strains with the acronym MF are from the pure culture collection of the All-Russian Institute of Plant Protection (VIZR, Laboratory of Mycology and Phytopathology), St. Petersburg, Russia; EGS from the personal collection of Dr. E. G. Simmons, Crawfordsville, IN, USA; CBS from the Culture collection of the Westerdijk Fungal Biodiversity Institute, Utrecht, The Netherlands; ATCC from the American Type Culture Collection, Manassas, Virginia, USA; and IRAN from the Fungal Culture Collections of the Iranian Research Institute of Plant Protection, Tehran, Iran. ** T—ex-type strain, R—representative strain. *** GenBank accession numbers highlighted in bold indicate sequences obtained during this study.

## Data Availability

Data is contained within the article.

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
