# Peer review of "Distinction of Alternaria Sect. Pseudoalternaria Strains among Other Alternaria Fungi from Cereals"

_jof, 2022, doi:10.3390/jof8050423_

Round 1
Reviewer 1 Report
In their paper, Gannibal et al. present the design of a PCR identification test for Alternaria spp from section Pseudoalternaria. In parallel, a formerly described PCR method to identify members of section Infectoriae was used. The specificity of the corresponding primer pairs was checked on reference strains from 10 different sections of the genus Alternaria and one strain belonging to the monophyletic lineage A. brassicae. Then a collection of 35 strains isolated in Russia from cereals was screened to discriminate strains of sections Infectoriae and Pseudoalternaria that could not be differentiated based solely on morphological criteria. Five strains corresponded to members of section Pseudoalternaria and were assigned based on a multilocus phylogeny to two species, A. kordkuyana and A. rosae. The authors also showed that these strains and most of the strains belonging to section Infectoriae did not produce the mycotoxin AOH.
The manuscript is clearly written and the presented results are convincing and may be useful for a quick and simple assignation of “Infectoria-like” isolates to section Pseudoalternaria or section Infectoriae.
`
Minor remarks :
-lane 14: replace “is” by “are”
-lane 174: I do not understand why the authors mention the “pathogenicity of the strains” as no results of pathogenicity tests are presented
Reference 1: delete 0 before Rotem
Reference 68 : Alternariol instead of Alternaroi
Author Response
Dear Reviewer,
The input from your review was helpful and greatly appreciated. The current version has undergone some revisions taking into account most of matters raised by three reviewers. The main changes are:
First reviewer comments
-lane 14: replace “is” by “are”
-lane 174: I do not understand why the authors mention the “pathogenicity of the strains” as no results of pathogenicity tests are presented
-Reference 1: delete 0 before Rotem
-Reference 68 : Alternariol instead of Alternaroi
RESPONSE: We thank the reviewer and followed all his/her four suggestions and have made necessary changes according to your indications.
Second reviewer comments
1) please add the morphological characters of genus Alternaria as "this genus is characterized by........ ";
RESPONSE: A brief morphological description was added in the second paragraph of Introduction: “This genus is characterized by dark-colored multicelled conidia with transverse and longitudinal septa. Conidia commonly occur in chains or sometimes remain solitary and usually contain an apical beak or tapering apical cells.”
2) For the BI analyses, add the soft MrModeltest 2.3, and was it used to determine the best-fit evolution model for each data set? How about is Four Markov chains were runing?
RESPONSE: The best nucleotide substitution model used for building the ML trees was determined in MEGA X 10.1. Markov chain Monte Carlo (MCMC) sampling method was used for calculation of Bayesian probability. We have added the detailed information on phylogenetic analysis in subsection 2.3 of Materials and Methods:
“Alignment of the sequences obtained for each strain was performed using Mega X 10.1 program [38]. Basic Local Alignment Search Tool (BLAST) was used to perform similarity search, by comparing the consensus sequences with sequences in NCBI GenBank database. The closest matching sequences were added to the alignment (Table 2). Phylogenetic analysis of combined sequences consisted of maximum likelihood (ML) and maximum parsimony (MP) performed with Mega X 10.1. ML analysis was completed on a neighbor joining starting tree generated automatically. Nearest neighbor interchange was used as the heuristic method for tree inference. The best nucleotide substitution model used for building the ML trees (TN93+G) was also determined in MEGA X 10.1. MP analysis was performed using the heuristic search option with 100 random taxon additions and subtree pruning regrafting method as the branch-swapping algorithm. All characters were unordered and of equal weight, and gaps were treated as missing values. Input parameter "maxtrees" was set to 100 and branches of zero length were collapsed. Bootstrap supports values for ML and MP trees branches were calculated with 1000 replicates. Additionally Bayesian probability (BP) calculation was done with Mr. Bayes v. 3.2.1. in Armadillo v. 1.1 [39]. using a Markov chain Monte Carlo (MCMC) sampling method. The general time-reversible model of evolution, including estimation of invariable sites and assuming a gamma distribution with six rate categories was used for Bayesian inference analyses. Four MCMC chains were run simultaneously, starting from random trees for 1000 generations and sampled every tenth generation for a total of 10 000 trees. Sequence data, aligment and phylogenetic trees were deposited in GenBank (MW478365-MW478374).”
3) please add the Aligned dataset was deposited in TreeBase (submission ID XXXXX);
RESPONSE: Thank you for the suggestion. We tried to deposit the dataset in TreeBase, however currently it is impossible to do. We enclose the file prepared for TreeBase to this submission.
4) in your study, is the Bayesian analysis resulted in a similar or same topology to MP analysis, and Bayesian analysis?? How much is an average standard deviation of split frequencies =?, and the effective sample size (ESS) =?
RESPONSE: The topology (composition of clusters) of trees resulted in Bayesian and MP analysis was the same. Now we use this word. The Bayesian analysis ran 1000 generations before the average standard deviation for split frequencies reached below 0.01 (0.009647). We would also like to draw attention to the fact that the tasks of the work did not include a detailed and large-scale reconstruction of phylogeny and revision of taxonomy. The main task was to identify the strains.
5) the author have to add a ITS tree to support you other markers tree in a larger scale on the family level (nLSU) or genus level;
RESPONSE: The low informativeness of ITS region for distinguishing of Alternaria species is well known. Previously the insufficiency of single ITS region analysis to distinguish even Alternaria sections was demonstrated (Woudenberg et al., 2013). Whereas the good resolution of gpd and ATP genes for distinguishing the species from sections Infectoriae and Pseudoalternaria was established and these regions often used in Alternaria phylogenetic study (Lawrence et al., 2014; Deng et al., 2018; Poursafar et al., 2018, 2019). In our study the bootstrap support of phylogenetic clade was high for confidence in species identification. Also we are confident that we studied strains belonging to Alternaria sect. Pseudoalternaria due to preliminary PCR diagnostics with specific primers designed in this study. Thus we believe that presentation of a larger scale phylogenetic ITS tree is superfluous.
6) the picture (Figure 2.) is blurry, please add a hand drawing picture to readers;
RESPONSE: We tried to make the image of conidia more sharpy. Probably when original file will be used to build the layout it looks better.
Comments from pdf-file:
Line 48. add a paragraph for the phylogenetic study of 1) this genus with this family and 2) the inside this genus and 3) among the sections.
RESPONSE: The aim of our study was identification of strains belonging to one section. Here we do not pretend on new data on transformation of family/genus/section phylogeny and taxonomy. There are 28 sections in Alternaria. Description of genus phylogeny could be short and uninformative or very long (if complete) with low correspondance to the purpose of the study. Thus we feel such paragraphs are superfluous.
Line 87. add the city, province and Country
RESPONSE: Thank you, information on country was added. St.Petersburg has a status of province and city at the same time.
Line 107. add a column for the references of the original sequences coming from
RESPONSE: We followed this recommendation and added requested information in Table 2.
Line 142. about ML, add the model which is employed; for MP, Trees were inferred using the heuristic search option with TBR branch swapping and 1,000 random sequence additions? OR others? For the BI analyses, MrModeltest 2.3 [30] was used to determine the best-fit evolution model for each data set? How about is Four Markov chains were runing? How much the random starting trees for generations? all the parameters are important for your phylogenetic analysis, it will suppot your study if you add them.
RESPONSE: Thank you. We answered this question above and added more detailed information on phylogenetic analysis in Materials and Methods.
Line 103. the outgroup is important for the tree, so selecting it as an outgroup for phylogenetic analyses of your dataset will give the reason or references.
RESPONSE: Thank you for this suggestion. We have included A. solani and A. alternata species as an outgroup in the phylogenetic tree. The sect. Porri and Alternaria that include these species are known to be distanced from the sect. Pseudoalternaria, while sect. Infectoriae is the sister group (Woudenberg et al., 2013). The phylogenetic relationships between Alternaria sections have been established earlier in large-scale studies (Lawrence et al., 2013; Woudenberg et al., 2013).
Line 227. Please add a macroscopical pictures for the Alternaria and cereals.
RESPONSE: Macroscopical pictures of Alternaria colonies are given in figure 2. We have no pictures of cereals. Strains were mainly isolated from symptomless seeds that make photographs not interesting.
Also we thank reviewer #2 for pointing out the flaws in the references list. It was corrected.
Third reviewer comments
(1) You should reorganize the Abstract to highlight the valuable contributions of this research work, including the designed primers APsF, APsR for distinction of sect. Pseudoalternaria from other Alternaria rather than the previously designed primers Ain3F and Ain4R, etc; too many keywords in the abstract;
RESPONSE: We thank reviewer for detailed comments. We have made necessary changes accordingly to your indications. The abstract was reorganized and supplemented. Keywords that repeat those in abstract were deleted.
(2) one important result for this manuscript is the PCR amplifications using designed primers APsF/APsR comparing with the previously designed primers Ain3F/Ain4R. However, authors presented these data in Table 1 in Methods, instead, in Results. Authors cited Table 1 only in line234, where is to present the toxin-producing.
RESPONSE: Thank you. We believe that one table combined all the data allows a better presentation of the results. Dividing it into several tables (separately information about analysed strains in Materials and Methods, a table with the results of PCR diagnostics, a table with toxin-producing ability) will be cumbersome due to the number of analyzed strains.
Minor revisions:
Line 37, need a coma after characteristicsï¼›
Line 63, need a full name for ITSï¼›
Line 181, should be Table 1 rather than Table 2;
Line 194, should be adjusted rather than ajusted;
Line 373, 0Rotem, J. should be Rotem?
Table 2, host plant for A. kordkuyana MF P094121shoule be Triticum aestivum Tricicum aestivum
RESPONSE: Thank you for the comments. All necessary corrections were made.
Table 1, you doubled strain A. kordkuyana MF P094121, you included strain MF P455041 in both sec. A. alternata and sec. A. arborescens; what are the meanings for those blanks for AOH content?
RESPONSE: Thank you for the comments. All points were corrected. We are sorry for technical errors in Table 1 which appeared during the final formatting. Two A. arborescens strains were included in specific PCR analysis and did not give a positive result in reactions with both analyzed primers sets. Their AOH-producing ability has not been assessed.
Line 236, You said all strains belonging to sect. Pseudoalternaria (5 strains), which five stains? MF P337011 also is zero?
RESPONSE: Thank you for the comment. Five A. sect. Pseudoalternaria strains were included in analysis of AOH-producing ability: four strains which were identified in this study (3 strains A. kordkuyana and 1 strain A. rosae) and one type strain of A. parvicaespitosa MF P472011 which was taken from the laboratory collection. The AOH-producing ability of strain MF P337011 has not been assessed.
Reviewer 2 Report
Dear Authors,
This paper is focusing on the Distinction of Alternaria sect. Pseudoalternaria strains among 2
other Alternaria fungi from cereals, which is to to incontrovertibly identify Russian A. sect. Pseudo- 81
alternaria strains by molecular phylogenetic, PCR and morphological analyses, and to 82
define their ability to produce the alternariol mycotoxin. It can be accepted after major revision.
Some comments on the text.
1) please add the morphological characters of genus Alternaria as "this genus is characterized by........ ";
2) For the BI analyses, add the soft MrModeltest 2.3, and was it used to determine the best-fit evolution model for each data set? How about is Four Markov chains were runing?
3) please add the Aligned dataset was deposited in TreeBase (submission ID XXXXX);
4) in your study, is the Bayesian analysis resulted in a similar or same topology to MP analysis, and Bayesian analysis?? How much is an average standard deviation of split frequencies =?, and the effective sample size (ESS) =?
5) the author have to add a ITS tree to support you other markers tree in a larger scale on the family level (nLSU) or genus level;
6) the picture (Figure 2.) is blurry, please add a hand drawing picture to readers;
Kind Regards,

Author Response

(The authors gave the same response as above.)

Reviewer 3 Report
Authors developed a sect. Pseudoalternaria-specific PCR method. The results are interesting and valuable to effectively distinct two phylogenetically and morphologically close Alternaria sections: the relatively well-known Infectoriae and the rarely-mentioned Pseudoalternaria. However, the major problems for current manuscript are organization and logistics, for example, (1) You should reorganize the Abstract to highlight the valuable contributions of this research work, including the designed primers APsF, APsR for distinction of sect. Pseudoalternaria from other Alternaria rather than the previously designed primers Ain3F and Ain4R, etc; too many keywords in the abstract; (2) one important result for this manuscript is the PCR amplifications using designed primers APsF/APsR comparing with the previously designed primers Ain3F/Ain4R. However, authors presented these data in Table 1 in Methods, instead, in Results. Authors cited Table 1 only in line234, where is to present the toxin-producing.
Minor revisions:
Line 37, need a coma after characteristicsï¼›
Line 63, need a full name for ITSï¼›
Line 181, should be Table 1 rather than Table 2;
Line 194, should be adjusted rather than ajusted;
Line 236, You said all strains belonging to sect. Pseudoalternaria (5 strains), which five stains? MF P337011 also is zero?
Line 373, 0Rotem, J. should be Rotem?
Table 1, you doubled strain A. kordkuyana MF P094121; you included strain MF P455041 in both sec. A. alternata and sec. A. arborescens; what are the meanings for those blanks for AOH content?
Table 2, host plant for A. kordkuyana MF P094121shoule be Triticum aestivum Tricicum aestivum
Author Response

(The authors gave the same response as above.)

Round 2
Reviewer 2 Report
Dear Author,
It is good for this presnet version, but it also needs some corrections.
1 please add the detailed morphological characters of genus Alternaria as "this genus is characterized by........ "; and you have to add the improtant reference;
2 the phylegenetic tree is simple, the author have to add 1) the full for the two different genus; 2) add all strains of your study in this tree to the author; 3) add the type sequences for different taxa; 4) add related morphological character in this tree to reflect the phylogeny;
Kind Regards,
3 add a key for the genus and you focusing secetion;
Author Response
Dear Reviewer,
Thank you very much for your comments. The current version has undergone some revisions. The changes are following:
- please add the detailed morphological characters of genus Alternaria as "this genus is characterized by........ "; and you have to add the improtant reference;
RESPONSE. The general morphological characters of genus Alternaria have been already added. Now we put the reference (#5) after description. We actually feel that detailed characters are not relevant for this article. Alternaria is extremely common genus, it is known for many mycologists and it was described for huge number of times. In our opinion it would be strange to give well-known information. However we added the section description: “Species of Alternaria sect. Pseudoalternaria form primary conidiophores aggregated on agar surface or developing from aerial hyphae, simple or branched with single apical pore []. Conidia are relatively small (usually no larger than 32 × 10 μm), ellipsoid to obclavate, medium brown to golden brown, mostly combined in short chains. Conidia form 3–4 transverse and 1–2 longitudinal septa. Sometimes conidia produce short to long, simple to multi-geniculate secondary conidiophores obtaining one to many conidiogenous loci.”
- the phylegenetic tree is simple, the author have to add 1) the full for the two different genus; 2) add all strains of your study in this tree to the author; 3) add the type sequences for different taxa; 4) add related morphological character in this tree to reflect the phylogeny;
RESPONSE. Thank you for the comment. We made phylogenetic tree (fig. 1) a bit larger. More type strains were included in phylogenetic reconstruction. Consequently table 2 (Alternaria spp. sequences used for phylogenetic study and primer design), alignment results and reference numbers were corrected.
Now the species of another genus, Stemphylium, is used as outgroup.
We do not know how to add morphological character in this tree in the frame of our article.
New Treebase file is enclosed.
- add a key for the genus and you focusing secetion;
RESPONSE. Some sections of Alternaria genus are very large and include several decades of species each (e.g. Porri, Alternaria, Infectoriae). So the summarizing section descriptions contain overlapping morphological characters. All sections were described based on phylogeny and not always relation between phylogeny and morphology is clear in Alternaria. Thus it is impossible to develop an unambiguous key for sections.
We also would like to note that our text was carefully edited by native English language speaker, Ian Riley that is mentioned in the Acknowledgments section.
Kind regards